METHODS AND RESOURCES

# Quantitative proteome comparison of human hearts with those of model organisms

**Nora Linscheid**[1], **Alberto Santos**[2], **Pi Camilla Poulsen**[1], **Robert W. Mills**[1], **Kirstine Calloe**[3], **Ulrike Leurs**[1], **Johan Z. Ye**[1], **Christian Stolte**[4], **Morten B. Thomsen**[1], **Bo H. Bentzen**[1], **Pia R. Lundegaard**[1], **Morten S. Olesen**[1], **Lars J. Jensen**[2], **Jesper V. Olsen**[2], **Alicia Lundby**[1,2]*

**1** Department of Biomedical Sciences, Faculty of Health and Medical Sciences, University of Copenhagen, Copenhagen, Denmark, **2** The Novo Nordisk Foundation Center for Protein Research, Faculty of Health and Medical Sciences, University of Copenhagen, Copenhagen, Denmark, **3** Department of Veterinary Science, Faculty of Health and Medical Sciences, University of Copenhagen, Copenhagen, Denmark, **4** New York Genome Center, New York, New York, United States of America

* alicia.lundby@sund.ku.dk

**Data Availability Statement:** The mass spectrometry proteomics data have been deposited to the ProteomeXchange Consortium via the PRIDE partner repository with the dataset identifier PXD012636 (accessible through https://www.ebi.

## Abstract

Delineating human cardiac pathologies and their basic molecular mechanisms relies on research conducted in model organisms. Yet translating findings from preclinical models to humans present a significant challenge, in part due to differences in cardiac protein expression between humans and model organisms. Proteins immediately determine cellular function, yet their large-scale investigation in hearts has lagged behind those of genes and transcripts. Here, we set out to bridge this knowledge gap: By analyzing protein profiles in humans and commonly used model organisms across cardiac chambers, we determine their commonalities and regional differences. We analyzed cardiac tissue from each chamber of human, pig, horse, rat, mouse, and zebrafish in biological replicates. Using mass spectrometry–based proteomics workflows, we measured and evaluated the abundance of approximately 7,000 proteins in each species. The resulting knowledgebase of cardiac protein signatures is accessible through an online database: atlas.cardiacproteomics.com. Our combined analysis allows for quantitative evaluation of protein abundances across cardiac chambers, as well as comparisons of cardiac protein profiles across model organisms. Up to a quarter of proteins with differential abundances between atria and ventricles showed opposite chamber-specific enrichment between species; these included numerous proteins implicated in cardiac disease. The generated proteomics resource facilitates translational prospects of cardiac studies from model organisms to humans by comparisons of disease-linked protein networks across species.

## Introduction

Experimental studies in model systems are key in investigating molecular mechanisms of cardiac disease and in therapeutic biomarker discovery [1,2]. The presumption underlying any cardiac disease study conducted in a model organism is that the model adequately recapitulates

ac.uk/pride/archive) and project name 'The protein expression landscape of the heart across humans and model organisms'. The website containing all cardiac protein expression data across species is accessible under atlas.cardiacproteomics.com.

**Funding:** This work was supported by grants from the Carlsberg Foundation (CF17-0209, https://www.carlsbergfondet.dk/), The Danish Council for independent Research (DFF-4092-00045, https://dff.dk/), and The Novo Nordisk Foundation (NNF15OC0017586. https://novonordiskfonden.dk/) to AL. The Novo Nordisk Foundation Center for Protein Research is funded in part by a generous donation from the Novo Nordisk Foundation (NNF14CC0001, https://novonordiskfonden.dk/). The funders had no role in study design, data collection and analysis, decision to publish, or preparation of the manuscript.

**Competing interests:** The authors have declared that no competing interests exist.

**Abbreviations:** ACTC1, actin; ARVC, arrhythmogenic right ventricular cardiomyopathy; CAA, chloroacetamide; DBN1, drebrin; DCM, dilated cardiomyopathy; DMD, dystrophin; DSP, desmoplakin; HCM, hypertrophic cardiomyopathy; HPLC, high-pressure liquid chromatography; LA, left atrium; LAMA2, laminin; LC–MS/MS, liquid chromatography tandem mass spectrometry; LV, left ventricle; MTUS1, microtubule-associated tumor suppressor 1; NaF, sodium fluoride; NEBL, nebulette; NES, nestin; PCA, principal component analysis; PLEC, plectin; RA, right atrium; RP-HPLC, reverse-phase high-pressure liquid chromatography; RT, room temperature; RV, right ventricle; TAGLN, transgelin; TCEP, tris(2-carboxyethyl)phosphine; TFA, trifluoroacetic acid; TNNI3, troponin 1; TPM1, tropomyosin 1; VCAN, versican.

relevant human cardiac physiology. Priority should be given to species that are anatomically and pathophysiologically similar with regards to the target disease. In more than half of studies testing regenerative medicine in cardiovascular diseases, the pig was the animal of choice [3]. The pig has also successfully served as a model for cardiac arrhythmias. Large mammals are in general considered the best translational models [4,5], but for practical reasons, smaller mammals are often favored. Proteins carry out the majority of biological functions, and protein-level differences between organisms often explain why pharmacological interventions fail to translate from animals to humans [6], as, e.g., illustrated by statins [7]. At present, there is a fundamental lack of comparative studies of the molecular buildup of hearts across species, hindering translation of findings from preclinical models.

Our ability to choose appropriate model organisms is directly connected to species-specific knowledge on protein networks. Consequently, determining differences in cardiac protein profiles across chambers and model organisms will directly benefit cardiac study design. Advances in mass spectrometry–based proteomics have enabled increasingly comprehensive mappings of proteomes [8–10] and have contributed insights on the dynamic changes in cardiac diseases [11,12] to identify protein targets [13]. Focused proteomics efforts on cardiac tissue from humans have underscored major protein differences across cardiac chambers [14]. Studies analyzing hearts from various species have outlined particular differences in sarcomeric proteins [15] as well as species-specific pathways [16]. Here, we utilized mass spectrometry–based methods to assess species- and region-specific protein composition of cardiac tissues. For quantitative purposes, we focused on freshly collected biopsy samples [14]. We performed systematic analyses of cardiac proteomes across cardiac chambers in humans and 5 commonly used model organisms: pig (*Sus scrofa*), horse (*Equus caballus*), rat (*Rattus norvegicus*), mouse (*Mus musculus*), and zebrafish (*Danio rerio*, also known as Brachydanio rerio or zebra danio). For each species, we identified and quantified approximately 7,000 proteins and compared protein profiles across species with respect to cardiac function and mechanisms of disease. Up to a quarter of chamber-enriched proteins was found to be higher expressed in different chambers between models, reflecting functional differences in heart rate, metabolism, and contractility. Using the differential protein profiles, we show why structural studies of hypertrophic cardiomyopathy are difficult to perform in zebrafish, and we conclude that the best animal model for arrhythmogenic right ventricular cardiomyopathy (ARVC) is pig. These results illustrate how our proteomics resource provides important insights for choice of model organisms in studying disease pathogenesis. The resource on multispecies cardiac proteomes is available as an open-access database: atlas.cardiacproteomics.com, which we envision to facilitate experimental design and interpretation of results across species and increase the translational prospect of cardiac findings.

## Results

### Deep proteome profiling of cardiac chambers across 6 species

To define cardiac protein expression profiles, biopsies from each cardiac chamber from 3 individuals per species were analyzed (Fig 1A, S1 Fig). Specifically, biopsies from left atrium (LA), right atrium (RA), left ventricle (LV), and right ventricle (RV) were collected from 3 mammals in each group of *E. caballus* (horse), *S. scrofa* (pig), *R. norvegicus* (rat), and *M. musculus* (mouse). For *D. rerio* (zebrafish), atrium (A) and ventricle (V) were collected and pooled from 10 fish per sample to ensure sufficient tissue material. For *Homo sapiens* (humans), LA, RA, and LV biopsies were taken during mitral valve replacement surgery, where collection procedure via RA precluded the possibility to sample from RVs. All biopsies were snap frozen in liquid nitrogen immediately after collection and stored at −80°C until further processing.

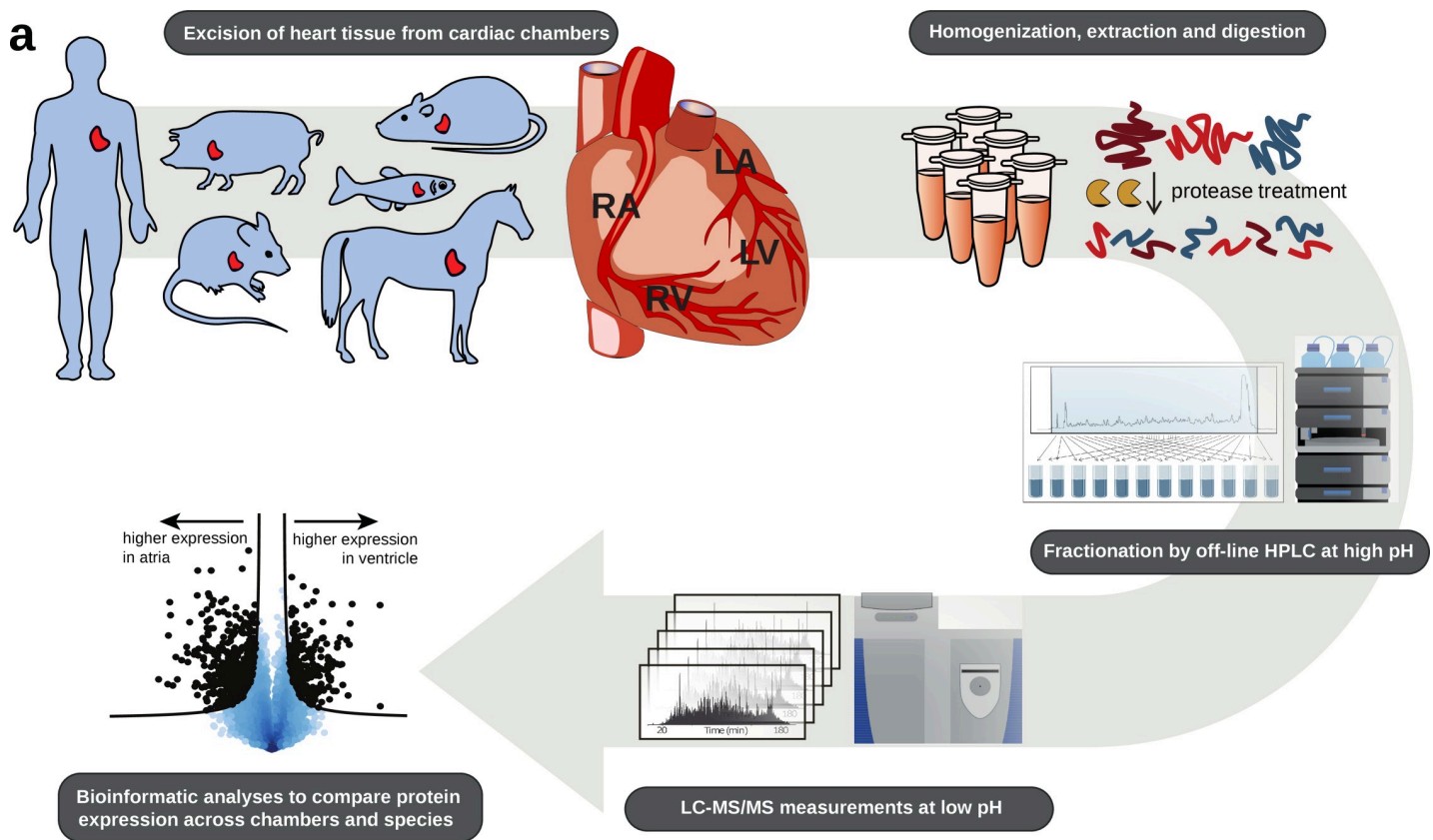

**b**

| Species | RV | LV | RA | LA | Total proteins |
|---|---|---|---|---|---|
| **Human** | n.d. | 6536 | 6624 | 6608 | 6729 |
| **Horse** | 6267 | 6181 | 6339 | 6274 | 6479 |
| **Pig** | 6907 | 6902 | 6971 | 7000 | 7177 |
| **Rat** | 7140 | 7133 | 7282 | 7282 | 7446 |
| **Mouse** | 6599 | 6522 | 6818 | 6784 | 6943 |
| **Zebrafish[1]** | 6865 | | 6988 | | 7158 |

[1] *Zebrafish have one atrium and one ventricle.*

*n.d. = not determined. RV tissue was not available from human patients.*

**Fig 1. Multispecies proteome mapping across cardiac chambers. (a)** Workflow for the determination of chamber-specific cardiac proteomes in human, horse, pig, rat, mouse, and zebrafish. Tissue biopsies were collected in triplicates. Biopsies were homogenized followed by protein extraction and digestion, desalted peptides were then fractionated, and the generated peptide fractions were analyzed by LC–MS/MS. Data were analyzed using MaxQuant and Perseus software. **(b)** Table summarizing the number of proteins measured in each species across cardiac chambers. HPLC, high-pressure liquid chromatography; LA, left atrium; LC–MS/MS, liquid chromatography tandem mass spectrometry; LV, left ventricle; RA, right atrium; RV, right ventricle.

Biopsies were homogenized using a ceramic bead mill, and proteins were extracted with a detergent-based buffer, which solubilizes cellular membranes and compartments [14,17]. Protein extracts were digested into peptides and pre-fractionated at high pH by reverse-phase high-pressure liquid chromatography (RP-HPLC) before mass spectrometry (liquid

chromatography tandem mass spectrometry, LC–MS/MS) analysis on a high-resolution Q-Exactive HF quadrupole Orbitrap tandem mass spectrometer [18]. In total, the study covers 654 LC–MS runs amounting to over 40 days of MS measurement time. All raw data files are made available via the Pride repository (see Data Availability). In Andromeda data search, only canonical protein sequences were included as global isoform-specific quantification cannot be done accurately by label-free approaches. Despite restraining our search to canonical protein sequences, we measured approximately 7,000 proteins in each cardiac chamber for each species (Fig 1B, S1–S6 Tables). Evaluation of acquired data is presented in S2–S8 Figs. For each species, we found high correlation between the 3 biological replicates with Pearson correlation coefficients mostly above 0.9. Principal component analyses (PCAs) showed that in general, variance between samples stemmed from differences between cardiac chambers (S3–S8 Figs). The quantitative proteomics dataset acquired represents a comprehensive mapping of cardiac protein expression profiles across chambers for human heart and 5 commonly used model organisms in cardiac research (S1C and S2 Figs).

## Building a database of protein profiles for all cardiac proteins across species

We built an open-data knowledgebase of cardiac protein profiles providing easy and efficient access to high-quality data: atlas.cardiacproteomics.com. The web page allows straightforward comparison of any identified protein in the dataset to all its homologs across species. It combines an easy-to-use web interface with intuitive data illustration capabilities (illustrated in Fig 2). Any protein can be queried in the online database; the output returns information on protein expression levels of all orthologs and paralogs across chambers for all analyzed species. Specific proteins of interest can thus easily be queried, and protein abundance across species and chambers can be evaluated without the need for any data handling. This allows individuals to make informed decisions on target proteins, model organisms, and experimental design.

Comparing protein expression profiles across species is not trivial since speciation events have led to a multitude of orthologous and paralogous genes that need to be mapped to their closest relatives. Since protein homology often creates one-to-many or many-to-one relationships of protein evolution across species, we created a database format that allows to contain these relationships fully and nonredundantly. To this end, we performed protein orthology/paralogy mapping based on EggNOG fine-grained orthology groups [19,20], which allowed us to preserve the full information available from our dataset. To make protein expression differences comparable across species, all raw data were first normalized to a common scale (S9 Fig, S7 Table), and protein abundance representations were translated from MS-based intensities into a confidence score (S10 Fig) [21]. This protein orthology network contained a total of 34,241 proteins connected by 294,850 binary relationships.

## Evolutionary conserved cardiac protein profiles

The quantitative proteomics datasets presented here allow global comparisons of cardiac protein profiles across species. Based on categorizing proteins into orthologous groups using Egg-NOG, as explained above, all homolog relations could be retained in the database. For visualization in this paper, we created a two-dimensional dataset by retaining 1 protein for each species per ortholog group based on highest degree of homology. Unsupervised hierarchical clustering on the resulting ortholog groups (Fig 3A, S8 Table) showed that (i) samples from each species form a cluster; and (ii) atria and ventricles form separate clusters. Notably, the species branches clustered according to evolutionary distance, with horse and pig, as well as mouse and rat forming common clusters on species level. As the most distant evolutionary

## a  Nppa tissues

Nppa [ENSMUSP00000099520]

Natriuretic peptide type A; Hormone playing a key role in cardiovascular homeostasis through regulation of natriuresis, diuresis, and vasodilation. Also plays a role in female pregnancy by promoting trophoblast invasion and spiral artery remodeling in uterus. Specifically binds and stimulates the cGMP production of the NPR1 receptor. Binds the clearance receptor NPR3.

Synonyms: Nppa, NPPA, P05125, P05125p, mP05125 ...

Linkouts: STRING UniProt

**b**

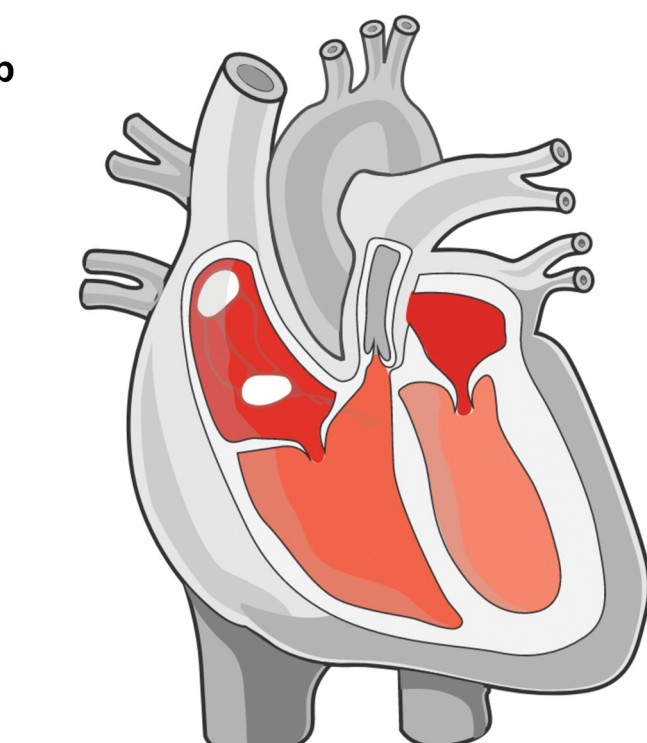

## c  Experiments

| Name | Source | Evidence |
| --- | --- | --- |
| Left atrium | Cardiac proteome | 1.4E10, 1.4E10, 4.8E9 |
| Right atrium | Cardiac proteome | 1.2E10, 7.6E9, 1.6E10 |
| Right ventricle | Cardiac proteome | 5.4E8, 9.2E8, 8.4E8 |
| Left ventricle | Cardiac proteome | 1.3E8, 1.4E8, 1.0E8 |

## d  Orthologs and paralogs

| Name | Organism | Homology | Correlation | Left atrium | Left ventricle | Right atrium | Right ventricle |
| --- | --- | --- | --- | --- | --- | --- | --- |
| Nppa | Rattus norvegicus gene | Ortholog | 0.99 | | | | |
| NPPA | Sus scrofa gene | Ortholog | 0.99 | | | | |
| NPPA | Equus caballus gene | Ortholog | 0.98 | | | | |
| NPPA | Homo sapiens gene | Ortholog | 0.83 | | | | |
| Nppa | Danio rerio gene | Ortholog | 0.23 | | | | |
| Nppb | Danio rerio gene | Ortholog | 0.17 | | | | |

**Fig 2. Website interface of cardiac protein expression database across species.** Example interface when searching for a protein of interest; example here is Nppa in mouse. **(a)** Detailed information of the queried protein as extracted from UniProt. A link for protein–protein interaction network of the protein as reported in STRING is provided. **(b)** Our measured protein expression across mouse heart chambers are displayed on a color scale in a graphic representation of the heart. **(c)** Table summarizing the measured experimental data. In this case, MS-based intensities were measured in all triplicates from all chambers. The measured protein intensity is provided in the "Evidence" column. Protein abundance is 2 orders of magnitude greater in the atria than in the ventricles. **(d)** All orthologs and paralogs identified in the dataset for Nppa are displayed in an adjacent table for comparison. In the database, measured protein intensities are translated into a multispecies confidence score for improved comparability. Nppa, natriuretic peptide type A.

species, the zebrafish had the largest vertical distance in the clustering. Cluster separation by cardiac chamber was particularly clear for smaller mammals and zebrafish, where even left and right sides of atria and ventricles clustered separately. This is likely a result of these animals being inbred strains and hence posing less molecular heterogeneity. The unsupervised hierarchical clustering underscores that our quantitative proteomics data reflect evolutionary relations between species.

To identify essential protein components of all hearts, we determined proteins that exhibited similar expression profiles across all species by ANOVA analysis and determined gene ontology enrichment. We found major overrepresentation of cytoplasmic and mitochondrial proteins, as well as proteins involved in translation and metabolic processes (Fig 3B, S8 Table). These findings are in line with previous reports [16] and underscore essential characteristics of the heart, such as its high energy demand. Among clusters that were significantly different between species, we found highest enrichment of cytoplasmic, vesicular, and mitochondrial proteins, proteins involved in binding and localization, RNA, peptide, and small molecule metabolic and catabolic pathways, as well as proteins with structural molecule activity (Fig 3B).

## Species-specific differences in heart proteins

We used PCA to define proteins driving differences across cardiac chambers and across species. Most of the variance in the dataset was explained by differences in expression between zebrafish, large mammals, and small mammals, which formed separate groups along principal component 1 and 2 (Fig 4A, upper panel). Prominent proteins driving this differentiation included NPPA, MYL7, MYL4, MYH11, SLC8A1, and ATP2B1, as well as several other channels, myofilaments, and extracellular matrix proteins (Fig 4A, lower panel). To facilitate the faster heart rate, the mouse and rat myocardium need to contract and relax much faster than is the case for the larger mammals. This is reflected in our data as reduced abundances of the slow-twitch myosin heavy chain MYH7 in mouse and rat compared to human, pig, and horse. Similarly, with the faster heart rate, the mouse and the rat are expected to have greater $Ca^{2+}$ handling capacities in the sarcoplasmic reticulum than the larger animals. And indeed, we observe greater abundances for the main calcium handling proteins RYR2, ATP2A2, and CASQ2 in mouse and rat compared to human, pig, and horse. Thus, essential molecular elements of fundamental cardiac functions are among the most differentially expressed proteins across species.

To explore which functional groups of cardiac proteins are differentially regulated between species, we performed ontology enrichment analyses on protein clusters which were significantly different between species (S11 Fig). Proteins that showed higher expression in small mammals (mouse and rat) compared to large mammals and zebrafish were enriched for mitochondrial proteins as well as proteins involved in ligation, translation, and peptide biosynthesis. Proteins that conversely showed lower expression in small mammals compared to other species were enriched for cellular amino acid metabolism. Taken together, this may indicate

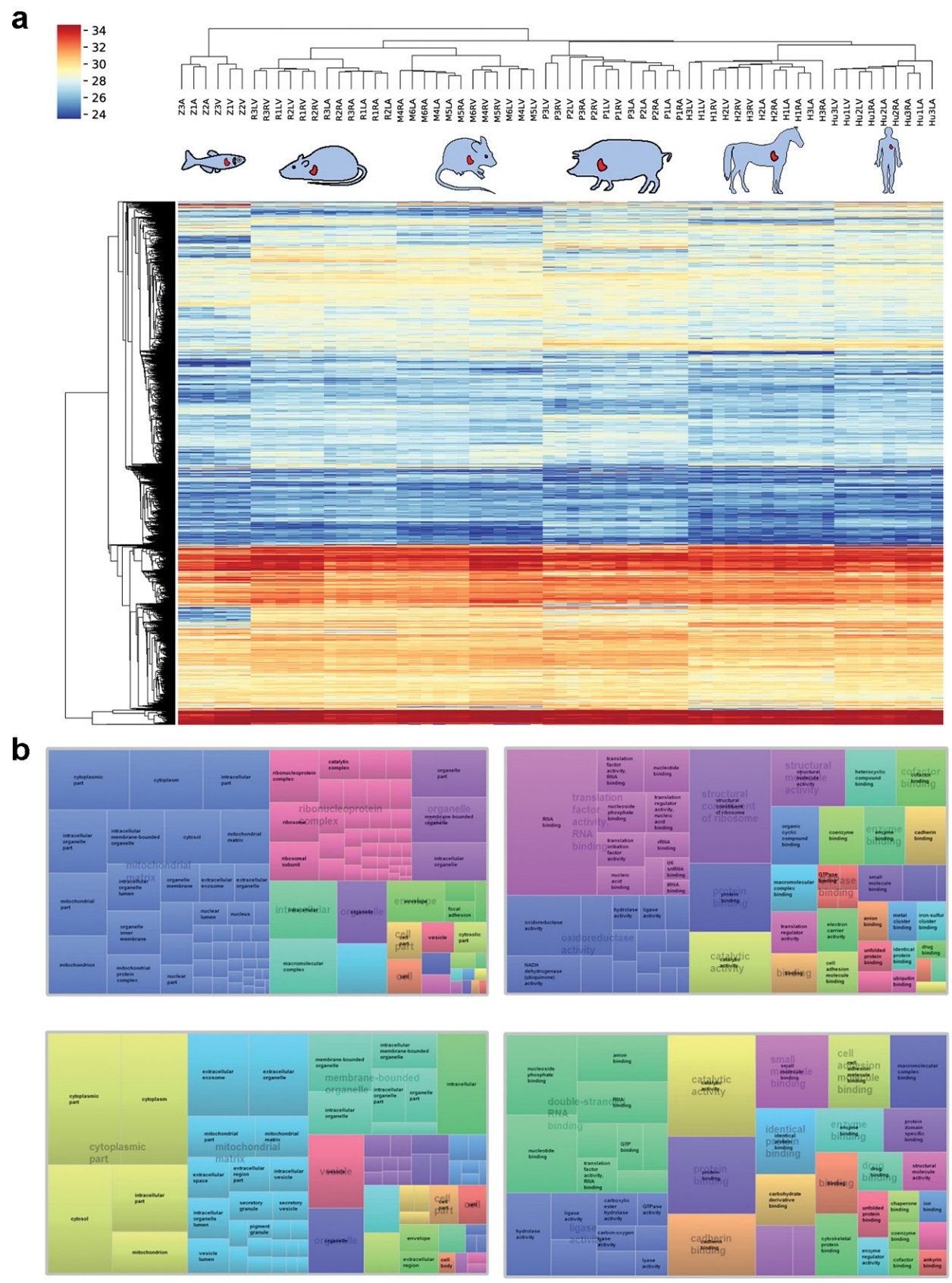

**Fig 3. Protein abundance profiles across species. (a)** Unsupervised hierarchical clustering of normalized protein intensities, for proteins measured in all samples resulted in grouping of samples from the same organism and reflects evolutionary distance between species as well as specific similarities and differences in protein expression. Proteins are colored by intensity with red showing highest and blue showing lowest intensity values (color bar denotes log2-transformed normalized protein intensities). **(b)** Visual representation of GO enrichment analysis of proteins with significantly different (upper panel) or similar (lower panel) abundance profiles across all species. Shown are representative enriched terms for GO, BP, CC, and MF, as well as KEGG pathways. Sizes of boxes are proportional to −log10 (p-value) of the enrichment (the larger, the more significant), and numbers denote the number of proteins enriched in the respective category. BP, biological process; CC, cellular component; GO, gene ontology; KEGG, Kyoto Encyclopedia of Genes and Genomes; MF, molecular function.

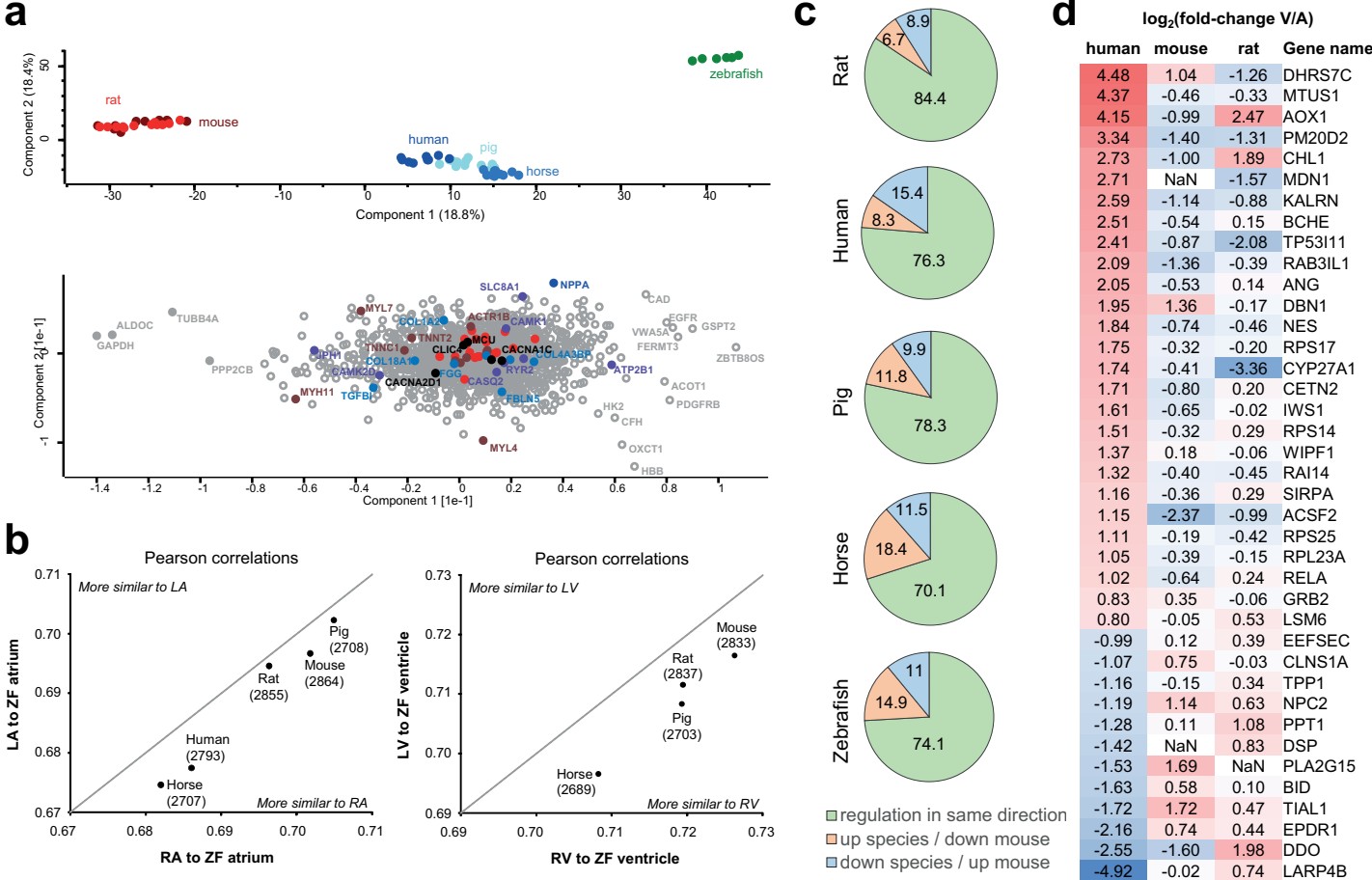

**Fig 4. Species-specific differences in protein abundance profiles. (a)** PCA (top) shows that the main sources of variation in the dataset are contributed by (i) the zebrafish samples (green) being different from all mammalian samples; and (ii) small mammals (red) being different from large mammals (blue). Analysis of which proteins explain most of the sample variance between samples (bottom) highlights, e.g., MYL4 and MYL7 showing high variance between zebrafish and large mammals along component 2, as well as NPPA, MYH11, and SLC8A1 highlighted as major contributors to the intermammalian differences. Reference sets of mitochondrial proteins are highlighted in orange, which show comparably lower loadings. This analysis is based on data presented in S8 Table. **(b)** Analysis of zebrafish protein abundance profile in atria and ventricle compared to corresponding mammalian protein abundance profiles in left and RV and atria. Pearson correlation analysis across all proteins consistently shows higher similarity to right side of the mammalian hearts, for ventricle as well as atria. This analysis is based on data presented in S8 Table. **(c)** Comparison of protein abundance differences between atria and ventricles across species. Protein abundances in ventricle compared to atria were calculated for mouse heart, and ratios were compared to all other species. Pie charts illustrate the percentage of proteins showing same direction of regulation (green), higher abundance in atria in other species in contrast to mouse (orange), and higher abundance in ventricle in other species contrary to finding in mouse (blue). **(d)** Proteins significantly differentially expressed between human ventricle and atria, which show opposite abundance profile in mouse and/or rat compared to human. log2 fold change of atria versus ventricle are shown, i.e., proteins higher expressed in ventricle are denoted positive (red), and those higher expressed in atria are denoted negative (blue). LA, left atrium; PCA, principal component analysis; RA, right atrium; RV, right ventricle.

differences in energy metabolism in hearts of small rodents. Several transcription factors were also enriched in both sets of clusters (ZF5, E2F-3, HES-7, and Sp1), potentially indicating differential regulation of downstream proteins.

Proteins differentially expressed in human compared to all other species (S12 Fig) were enriched for sarcolemma, structural constituent of muscle, sphingolipid pathway, and voltage-gated calcium channel activity. Specifically, ATP1A1, MYH11, JPH1, CACNA1C, CACNB2, and CAMK1 were among the significantly different proteins in human, highlighting that proteins fundamental to cardiac function can be some of the most differentially expressed compared to model organisms. Other proteins with a unique protein profile for the human heart include versican (VCAN), an extracellular proteoglycan involved in heart development; plectin (PLEC), a cytoskeletal linker found in nearly all mammalian cells; and transgelin (TAGLN), an actin binding protein involved in Ca-independent smooth muscle contraction.

Zebrafish is a popular model organism in cardiac research, although its physiology with only 2 cardiac chambers is markedly different from mammals. We examined which side of the mammalian heart the 2-chambered zebrafish heart resembles most with regard to its molecular profile. Our analyses consistently showed greatest similarity between zebrafish heart and the right half of mammalian hearts (Fig 4B, S13 Fig). This was the case for atria as well as ventricle. We propose this to reflect the zebrafish circulatory system being a low-pressure system, and hence the function of the heart resembling the right side of mammalian hearts serving the lower-pressure pulmonary system.

Lastly, we compared differentially expressed proteins between atria and ventricles across all species. For each species, we computed protein expression fold change between atria and ventricles and determined significance of differential expression by 2-sample *t* test. We compared these significantly different proteins from each species to the respective fold change expression in mouse, as mouse data showed the highest degree of completeness in the EggNOG mapping. In this analysis, 20% to 25% showed opposite chamber-enriched regulation across species (Fig 4C, S10A and S10B Fig). These proteins with opposing expression patterns include proteins implicated in cardiac function and disease. For instance, we identified 39 proteins that were significantly overexpressed in human atria or ventricle but showed opposite expression patterns in mouse and/or rat (Fig 4D): These proteins included important desmosomal proteins such as desmoplakin (DSP), transcription factors such as NF-kappa-B (RELA), and cytoskeleton-modifying proteins such as microtubule-associated tumor suppressor 1 (MTUS1), drebrin (DBN1), and nestin (NES). We confirmed the chamber-enriched expression of these proteins by comparison against independent datasets (S14 and S15 Figs, S9 Table).

## Molecular assessment of model organisms for cardiac disease studies

In addition to the untargeted analyses above, we compared expression of proteins known to be involved in particular diseases. We compared left ventricular protein expression across all species for proteins involved in hypertrophic and dilated cardiomyopathy (HCM and DCM) [22] and performed unsupervised hierarchical clustering on those proteins (Fig 5A and 5B). Notable differences for the HCM-associated genes include lower expression of MYL2 and 3, ACTC1, and MYH7 in zebrafish in comparison to the other species, indicating that extra care has to be taken when translating study results from zebrafish to human for these particular proteins. In DCM, expression of cytoskeletal and contractile proteins such as tropomyosin 1 (TPM1), nebulette (NEBL), troponin 1 (TNNI3), laminin (LAMA2), dystrophin (DMD), and actin (ACTC1) was again lower in zebrafish in comparison to the other species. Considering the crucial functions of these proteins in cardiac muscle tissue, attention has to be paid when designing studies in zebrafish involving these proteins.

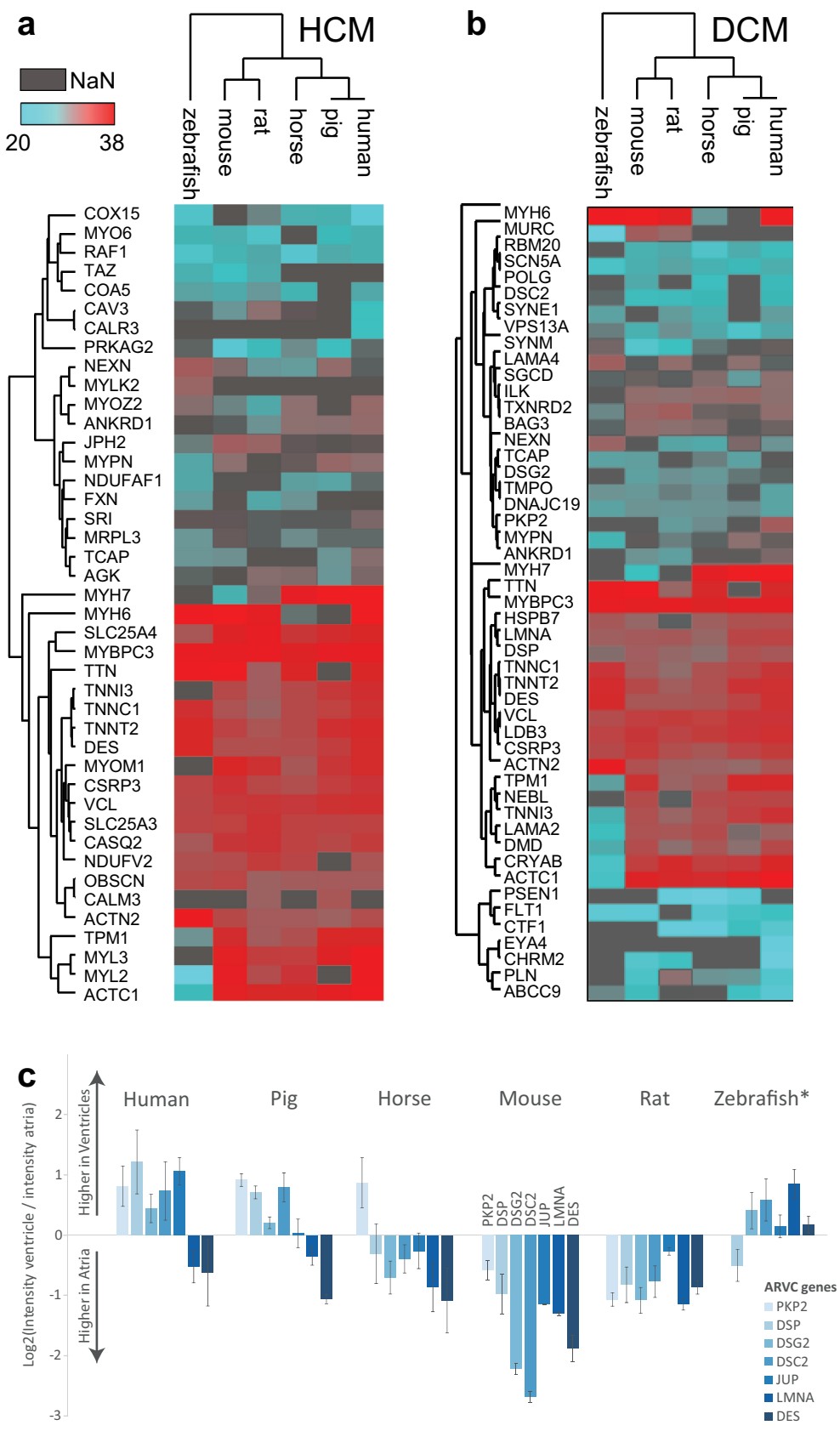

**Fig 5. Protein abundance profiles for cardiac disease–associated proteins across species. (a, b)** Median protein abundances in LVs are shown for proteins reported to be involved in HCM (panel a) and DCM (panel b) across species. Color scale represents log2-transformed protein intensities (red: highest abundance, turquoise: lowest abundance, and gray: not available). **(c)** Comparison of protein abundance ratios between atria and ventricle for proteins encoded by 7 genes involved in ARVC. log2 fold change between ventricle and atria are shown. Note that the human ratio profile is best reflected by pig, while profiles in other species differ markedly. Bar height denotes the sample mean, and error bars denote standard deviation. This analysis is based on data presented in S8 Table. ARVC, arrhythmogenic right ventricular cardiomyopathy; DCM, dilated cardiomyopathy; HCM, hypertrophic cardiomyopathy; LV, left ventricle.

Finally, we analyzed the expression pattern of the 7 proteins most commonly involved in ARVC across species [23,24]. We found that 5 of the primary ARVC-associated genes were more highly expressed in the ventricles of human, whereas expression patterns varied considerably among the other species, with the pig ARVC-associated protein expression profile being the most concordant with human (Fig 5C). Which model organism particularly well represents certain processes or diseases cannot be answered globally, but depends on the question asked. As an example to illustrate how our data can be of use in interpreting or designing studies performed in model organisms, we plotted profiles of protein expression in LV for HCM- and DCM-related genes, depicting how proteins differ in expression between model organisms and human (S16 Fig, S8 Table). Protein profiles are generally similar across organisms for both diseases, with some marked departures: For the HCM example, the largest expression differences are seen for MYPN, NEXN, and OBSCN expression between zebrafish and human. On the other hand, PRKAG2, SLC25A4, TNNC1, and TNNT2 are similarly abundant between zebrafish and human, while (sometimes strongly) lower abundant in all other mammals. While our data cannot always point to the best model organism for a given disease, it thus point to candidate genes and proteins which may cause differential responses to medication or disease progression between organisms. For a researcher working on specific processes or diseases, such analyses could help explain discrepancies between studies in different model organisms or prioritize a list of target proteins to attribute these differences to through follow-up studies.

## Discussion

In the current study, we utilized a proteomics approach to generate a high-resolution map of the cardiac protein landscape across humans and 5 commonly used model organisms allowing for interchamber and interspecies comparison of protein abundances. Our design focused on quantitative rather than qualitative information: This is essential for an ultimate push toward translational prospect of findings from basic research [25]. Cardiac disease progression is often characterized by protein remodeling [26], and as most studies are performed in preclinical models, it remains an important need to translate such findings to humans [27]. Despite tremendous amounts of genomics and transcriptomics datasets, corresponding information on cardiac proteomes and their differences across models is still scarce [28]. Previous proteomics studies of model organisms have illuminated portions of their cardiac proteomes [29,30], often with a particular focus such as cardiac development[31], disease models [32], subcellular protein expression [33,34], phosphorylation [35–37], protein turnover [12,38], or smaller mammals and amphibians [16]. The deepest human heart dataset obtained to date presents an impressive atlas [39], but its usefulness for quantitative comparison is limited as it was acquired from tissue collected several days postmortem [14]. Using the resource generated herein, we showed which protein profiles are shared and which differ across species providing chamber-specific, quantitative information of approximately 7,000 proteins expressed in hearts.

Comparison of protein expression across organisms is challenging [10], especially due to evolutionary distance. An important aspect is the completeness and correctness of employed

protein sequence and orthology databases, especially for less-studied organisms. These differences could potentially introduce a systematic bias to protein identification and data analysis. Here, we minimize such effects by employing the gene-centric protein database (Ensembl) and the corresponding protein orthology tree (EggNOG). This approach was chosen because the genomes of all investigated species have been sequenced, while knowledge on protein level is vastly different. Accordingly, the employed data-driven approaches, such as hierarchical clustering and similarity metrics, can yield new insights even when curated knowledge is sparse. As one example, the zebrafish is becoming an increasingly popular model organism in cardiac studies due to its versatile use in high-throughput drug screening, CRISPR technology [40], and the possibility to perform in vivo optical mapping of action potentials and calcium fluxes [41]. Querying our quantitative proteome data, we show that the zebrafish heart is generally more similar to the right side of the mammalian heart, likely a reflection of the zebrafish heart being a low-pressure system.

The analysis of proteins with differential atrial–ventricular abundances showed that up to a quarter had inverted protein differentials in other species, reflecting functional differences in heart rate, metabolism, contractility, as well as gene products for cardiomyopathies. Cardiomyopathy-related protein profiles illustrated how human profiles for ARVC were exclusively consistently recapitulated in pig hearts, a finding of immediate importance when translating preclinical data to humans. As demonstrated for dilated and hypertrophic cardiomyopathies, protein expression differences between species are complex and not predictable for a disease as a whole: The expression of each protein and its immediate network must be individually evaluated.

We propose that the ability of an animal model to recapitulate human heart disease states is linked to the similarity in relative abundance of protein networks relevant to the disease [17,42]. Herein, we present a quantitative dataset of cardiac protein expression across humans and common model organisms at cardiac chamber resolution, revealing molecular features that are shared among all species, as well as specific features that are species dependent, together assembling a portrait of cardiac protein signatures for all commonly used model organisms. Our results allow meaningful comparisons both between species as well as between cardiac chambers within a species, even when curated knowledge is sparse. An important next step will be to expand on this resource with information on the cellular composition across the cardiac regions [29] and how these differ across humans and model organisms [43] and from there expanding to evaluate protein abundances per cardiac cell type [44] as technologies improve and allow for it. We expect that this data, publicly accessible in database format, may aid in choosing the best-suited model organism to test a given hypothesis, as well as to evaluate findings from studies conducted in model organisms for human physiology.

## Materials and methods

A detailed Materials and methods section is provided in S1 Text.

### Materials

If not specified otherwise, chemicals and reagents were acquired from Sigma-Aldrich, United States of America. Chromatography solvents were acquired from VWR, USA.

### Tissue collection

We collected biopsies from LA, RA, LV, and RV in mammals and from atrium (A) and ventricle (V) in zebrafish. Human biopsies were collected during minimal invasive mitral valve replacement surgery via the RA, and due to the nature of this procedure, the RV was not

accessible and therefore not included. Collection of the human heart tissue was approved by the Ethics Committee of the Capital Region of Copenhagen (protocol reference number: 16238) and was in accordance with the Declaration of Helsinki. All animal experiments were performed according to the European Union legislation for protection of animals used for scientific experiments and was approved by the Danish National Animal Experiments Inspectorate (licenses 2012-14-2934-00041 and 2012-15-2934-00083). Due to differences in heart sizes, biopsies from human, pig, and horse were specifically taken from the muscular part of the free walls; for rat and mouse, entire free wall biopsies were collected, and for zebrafish, entire chambers were collected and pooled from 10 fish per sample. All biopsies were snap frozen in liquid nitrogen immediately after collection and stored at −80˚C until further processing.

## Tissue homogenization, digestion, and fractionation

Frozen tissue biopsies were homogenized on a Precellys24 homogenizer (Bertin Technologies, France) with ceramic beads (2.8 and 1.4 mm zirconium oxide beads, Precellys) in tissue incubation buffer (50 mM Tris-HCl, pH 8.5, 5 mM EDTA, 150 mM NaCl, 10 mM KCl, 1% Triton X-100, 5 mM sodium fluoride (NaF), 5 mM beta-glycerophosphate, 1 mM Na-orthovanadate, containing Roche complete protease inhibitor). After homogenization, samples were incubated for 2 hours at 4˚C (20 rpm). Samples were centrifuged (15,000x g, 20 minutes, 4˚C), and the soluble fraction was collected and protein precipitated using ice-cold acetone (25% final concentration, VWR) for 1 hour at −20˚C followed by centrifugation (400x g, 1.5 minutes). Supernatants were discarded and protein resuspended in Guanidine-HCl buffer (6M Gnd-HCl, 50 mM Tris-HCl, pH 8.5, 5 mM NaF, 5 mM beta-glycerophosphate, 1 mM Na-orthovanadate, containing Roche complete protease inhibitor, 5 mM Tris(2-carboxyethyl)phosphine (TCEP), 10 mM chloroacetamide (CAA)) and incubated in the dark at room temperature (RT) for 15 minutes. Protein was digested using endoproteinase Lys-C (Trichem ApS, Denmark; 1:100 w/w) for 1 hour, 750 rpm at 30˚C in the dark, followed by dilution (1:12 with 50 mM Tris-HCl, pH 8) and digestion with trypsin overnight (16 hours) at 750 rpm and 37˚C (Life Technologies, USA, 1:100 w/w). Digestions were quenched by addition of trifluoroacetic acid (TFA, 1% final concentration) and centrifuged (14,000x g, 10 minutes). Soluble fractions were desalted and concentrated on C18 SepPak columns (Waters, USA) according to manufacturer's protocol. Up to 1-mg peptide was fractionated by RP-HPLC on a Dionex UltiMate 3000 HPLC system (Thermo Fisher Scientific, USA) equipped with an XBridge BEH C18 Sentry Guard Cartridge pre-column (130 Å, 3.5 um particle size, 4.6*20 mm, Waters) coupled to an XBridge Peptide BEH C18 packed column (130 Å, 3.5 um particle size, 4.6*250 mm, Waters) at 1 mL/min flow rate. The following gradient elution program was used at a constant supply of 10% solvent C (25 mM ammonia, pH 10): 0 to 49 minutes: 10% to 25% solvent B (100% ACN) linear gradient, 50 to 54 minutes: 25% to 70% B linear gradient, and 55 to 59 minutes: 70% B isocratic flow, followed by column re-equilibration at 5% B for 10 minutes as previously described [12]. Peptides were collected from 0 to 60 minutes in 10 concatenated fractions. Fraction volume was reduced by vacuum centrifugation to 20 to 100 μL.

## LC–MS/MS measurements

Fractionated peptide samples were analyzed by online reversed-phase liquid chromatography coupled to a Q-Exactive Plus quadrupole Orbitrap tandem mass spectrometer (Thermo Fisher Scientific, Bremen, Germany). Peptide samples were separated on 15-cm fused silica emitter columns pulled and packed in-house with reversed-phase ReproSil-Pur C18-AQ 1.9um resin (Dr. Maisch GmbH, Ammerbuch-Entringen, Germany) in a 1-hour multistep linear gradient (0.1% FA constant; 2% to 25% ACN in 45 minutes, 25% to 45% ACN in 8 minutes, 45% to

80% ACN in 3 minutes) followed by a short column re-equilibration (80% to 5% ACN in 5 minutes, 5% ACN for 2 minutes).

Raw MS data were processed using the MaxQuant software (version 1.5.3.19, Max-Planck Institute of Biochemistry, Department of Proteomics and Signal Transduction, Munich, Germany), and proteins were identified with the built-in Andromeda search engine based on Ensembl [45] canonical protein collections for each species. False discovery rate cutoffs were set to 1% on peptide, protein, and site decoy level (default), only allowing high-quality identifications to pass. Because all raw intensities showed similar distributions, data were normalized across species by quantile normalization based on the Bioconductor R package limma [46].

We normalized the data globally across species by median centering and used the EggNOG database [20] to map orthologous groups of proteins between species. We then systematically compared similarities and differences in protein expression across species. Data analysis was performed using Perseus [47], Cytoscape [48], R, and Python. For representation in the database, the intensity values were translated into a multispecies confidence score by comparison to a gold standard as previously described [21]. See further details in the Supporting information section.

## Supporting information

**S1 Text. Document containing full Materials and methods section.**
(DOCX)

**S1 Table. All proteins identified in human heart tissue.** Proteomic investigation of human heart biopsies from RA, LA, and LV resulted in identification of 6,729 proteins. LA, left atria; LV, left ventricle; RA, right atria.
(XLSX)

**S2 Table. All proteins identified in mouse heart tissue.** Proteomic investigation of mouse heart biopsies from RA, LA, and LV resulted in identification of 6,943 proteins. LA, left atria; LV, left ventricle; RA, right atria.
(XLSX)

**S3 Table. All proteins identified in rat heart tissue.** Proteomic investigation of rat heart biopsies from RA, LA, RV, and LV resulted in identification of 7,446 proteins. LA, left atria; LV, left ventricle; RA, right atria; RV, right ventricle.
(XLSX)

**S4 Table. All proteins identified in pig heart tissue.** Proteomic investigation of pig heart biopsies from RA, LA, RV, and LV resulted in identification of 7,177 proteins. LA, left atria; LV, left ventricle; RA, right atria; RV, right ventricle.
(XLSX)

**S5 Table. All proteins identified in horse heart tissue.** Proteomic investigation of horse heart biopsies from RA, LA, RV, and LV resulted in identification of 6,479 proteins. LA, left atria; LV, left ventricle; RA, right atria; RV, right ventricle.
(XLSX)

**S6 Table. All proteins identified in zebrafish heart tissue.** Proteomic investigation of zebrafish samples from atria (A) and ventricle (V) resulted in identification of 7,158 proteins.
(XLSX)

**S7 Table. Protein intensity input file for database.** Protein intensities measured, with median subtraction for normalization, combined file for all species. Input file used in the online

database at atlas.cardiacproteomics.com.
(XLSX)

**S8 Table. Protein intensity matched between species based on EggNOG ortholog groups to compare protein profiles across species.** The file contains 2 tabs, one with proteins identified in all samples (with 100% valid values, input for hierarchical clustering Fig 3A) and one containing all proteins identified with at least 2 valid values per chamber (rest imputed, input for ANOVA analyses reported in Fig 3B and S11 and S12 Figs). Please note that this table contains far less proteins than the online database, since orthology groups had to be collapsed to a one-to-one mapping. For full orthology information, please visit the provided database website.
(XLSX)

**S9 Table. Proteome comparison with 2 previously published human heart proteomes.**
(XLSX)

**S1 Fig. Information on samples included in the study. (a)** Comparison of heart size and heart rate in horse, human, pig, rat, mouse, and zebrafish. The size of the mammal hearts is correlated with their physical size, while the heart rate is negatively correlated. The zebrafish deviates from this trend by having the smallest heart, and only half the heart rate of a rat. Due to differences in heart size, biopsies were collected in different fashions. In humans, biopsies were collected by needle biopsy during cardiac surgery. In horse and pig, biopsies were alike collected from the free walls of the myocardium. In rodents, whole free walls of cardiac chambers were collected. In zebrafish, 10 whole chambers were collected and pooled per sample. **(b)** Patient information for the 3 human individuals included in the study. Biopsies were collected from 3 males undergoing mitral valve replacement surgery. **(c)** The representation of the species studied herein for cardiovascular research in general was evaluated from the number of animals used in basic research as well as translational and applied research within cardiovascular research in the European Union in 2017. All numbers were obtained from the "2019 report on the statistics on the use of animals for scientific purposes in the Member States of the European Union in 2015–2017." For basic research in the relevant field, the species we studied herein cover 98.6% of the animals used, and for translational and applied research, the species studied account for 96% of the animals used. BMI, body mass index; CAD, coronary artery disease; NYHA, New York Heart Association functional classification; PVD, peripheral vascular disease, alcohol in units per week.
(TIF)

**S2 Fig. Overview over cardiac proteomic studies in organisms employed in this study.** This list does not claim completeness.
(TIF)

**S3 Fig. Evaluation of human proteome data. (a)** Density plots of intensities in log10-space displayed before (upper panel) and after quantile normalization (lower panel). Quantile normalization was performed to remove minor technical variation from the dataset. **(b)** Pearson correlations coefficients of log-transformed quantile normalized protein intensities across all samples. Hu1 through Hu3 denote the 3 human patients in the study, and RA, LA, and LV denote right atrium, left atrium, and left ventricle, respectively. **(c)** Overlap of identified proteins across heart chambers are shown in a Venn diagram. More than 95% of proteins were identified in all chambers. **(d)** PCA of samples shows clear distinction between atria and ventricle samples in the first principal component that explains 32.3% of the variance in the dataset. Analyses are based on data presented in S1 Table. LA, left atria; LV, left ventricle; PCA,

principal component analysis; RA, right atria.
(TIF)

**S4 Fig. Evaluation of mouse proteome data. (a)** Density plots of intensities in log10-space displayed before (upper panel) and after quantile normalization (lower panel). **(b)** Pearson correlation coefficients of log-transformed quantile normalized protein intensities across all samples. M1 through M3 denote the 3 mice in the study, and LA, RA, LV, and RV denote left atrium, right atrium, left ventricle, and right ventricle, respectively. **(c)** Overlap of identified proteins across heart chambers are shown in a Venn diagram. More than 92% of proteins were identified in all chambers. **(d)** PCA of samples shows clear distinction between atria and ventricle samples, and separation between right and left atria/ventricle in the first 2 principal components that explain 54.6% of the variance in the dataset. Analyses are based on data presented in S2 Table. LA, left atria; LV, left ventricle; PCA, principal component analysis; RA, right atria; RV, right ventricle.
(TIF)

**S5 Fig. Evaluation of rat proteome data. (a)** Density plots of intensities in log10-space displayed before (upper panel) and after quantile normalization (lower panel). **(b)** Pearson correlation coefficients of log-transformed quantile normalized protein intensities across all samples. R1 through R3 denote the 3 rats in the study, and LA, RA, LV, and RV denote left atrium, right atrium, left ventricle, and right ventricle, respectively. **(c)** Overlap of identified proteins across heart chambers are shown in a Venn diagram. More than 93% of proteins were identified in all chambers. **(d)** PCA of samples shows clear distinction between atria and ventricle samples in the first principal components that explains 33.6% of the variance in the dataset. Analyses are based on data presented in S3 Table. LA, left atria; LV, left ventricle; PCA, principal component analysis; RA, right atria; RV, right ventricle.
(TIF)

**S6 Fig. Evaluation of pig proteome data. (a)** Density plots of intensities in log10-space displayed before (upper panel) and after quantile normalization (lower panel). **(b)** Pearson correlation coefficients of log-transformed quantile normalized protein intensities across all samples. P1 through P3 denote the 3 pigs in the study, and LA, RA, LV, and RV denote left atrium, right atrium, left ventricle, and right ventricle, respectively. **(c)** Overlap of identified proteins across heart chambers are shown in a Venn diagram. More than 93% of proteins were identified in all chambers. **(d)** PCA of samples shows distinction between atria and ventricle samples in the first 2 principal components that explain 37.5% of the variance in the dataset. Analyses are based on data presented in S4 Table. LA, left atria; LV, left ventricle; PCA, principal component analysis; RA, right atria; RV, right ventricle.
(TIF)

**S7 Fig. Evaluation of horse proteome data. (a)** Density plots of intensities in log10-space displaying before (upper panel) and after quantile normalization (lower panel). **(b)** Pearson correlation coefficients of log-transformed quantile normalized protein intensities across all samples. H1 through H3 denote the 3 horses in the study, and LA, RA, LV, and RV denote left atrium, right atrium, left ventricle, and right ventricle, respectively. **(c)** Overlap of identified proteins across heart chambers are shown in a Venn diagram. More than 93% of proteins were identified in all chambers. **(d)** PCA of samples showing heart chambers or replicates along the first 2 principal components. The horse samples display greater dispersion in the PCA than observed in the other species tested, which is likely explained by age and strain differences between the 3 horses included in the study. Analyses are based on data presented in S5 Table. LA, left atria; LV, left ventricle; PCA, principal component analysis; RA, right atria; RV, right

ventricle.
(TIF)

**S8 Fig. Evaluation of zebrafish proteome data. (a)** Density plots of intensities in log10-space displayed before (upper panel) and after quantile normalization (lower panel). **(b)** Pearson correlation coefficients of log-transformed quantile normalized protein intensities across all samples. Z1 through Z3 denote the 3 zebrafish in the study, and A and V denote atrium and ventricle. **(c)** Overlap of identified proteins across heart chambers are shown in a Venn diagram. More than 97% of proteins were identified in all chambers. **(d)** PCA of samples shows clear distinction between atrial and ventricular samples in the first principal component that explains 47.4% of the variance in the dataset. Analyses are based on data presented in S6 Table. PCA, principal component analysis.
(TIF)

**S9 Fig. Protein intensity distributions of each species before and after data normalization. (a)** Raw intensity sample distributions from each species. **(b)** Overlaid distributions from a. **(c)** Overlaid normalized raw intensity distributions by median subtraction and centering at new median 1E8 shows good overlap of data samples after normalization. This indicates sufficient similarity for comparison across species. Analyses are based on data presented in a and b (S1–S6 Tables) and c (S7 Table).
(TIF)

**S10 Fig. Calculation of cardiac gold standard.** For representation in the database, the intensity values were translated into a multispecies confidence score by comparison to a gold standard as previously described [21]. We used the human dataset to make this comparison and calculate the new scoring scheme across species. To convert intensities into confidence scores, the agreement with the gold standard was quantified using fold enrichment. To calculate fold enrichment, we sorted proteins by intensity value and within sliding windows (window size = 50) calculated the fraction of proteins in the dataset found annotated to heart in the gold standard divided by the fraction expected when randomly sampling proteins from the gold standard. Then, we used the resulting curve (protein intensity, fold enrichment) to fit a function to translate intensities into confidence scores: $confidence\ score = a_0 + (a_1 - a_0)/(1 + e^{-a_2(x-a_3)})$, where $x$ is the mean intensity within a sliding window of 50 proteins. Analyses are based on data presented in S7 Table.
(TIF)

**S11 Fig. Significantly differentially expressed proteins across species. (a)** Subset of hierarchical clustering analysis on median protein intensities per chamber, showing all protein groups deemed significantly different based on multiple-sample ANOVA testing for differences between evolutionary groups of species (fish, small mammals, and large mammals) at false discovery rate of 0.01. Six clusters were selected that showed specific up- or down-regulation of protein groups in small mammals (rat and mouse) as compared to other species. **(b)** Profile plots for each cluster color-coded corresponding to panel a. Each line represents the intensity profile across samples of 1 orthologue group. Analyses are based on data presented in S8 Table.
(TIF)

**S12 Fig. Significantly different proteins in humans.** Subset of hierarchical clustering analysis on protein intensities across species, showing all protein groups deemed significantly different based on multiple-sample ANOVA testing for differences between human against all other

species at false discovery rate of 0.01.
(TIF)

**S13 Fig. Evaluation of zebrafish similarity to left-/right-side heart chambers.** Scatter plot of zebrafish heart chamber similarity to left/right heart chamber within mouse, rat, pig, horse, and human. Similarity measures cosine distance (a) and Euclidean distance (b) were used. The zebrafish atrium is more similar to the right atrium in all species except the pig; the zebrafish atrium is slightly more similar to the left pig atria by a very small margin. When comparing the zebrafish ventricle to the left/right ventricles of the mouse, rat, pig, and horse, the zebrafish displays more similarity to the right ventricle of every species. Analyses are based on data presented in S8 Table.
(TIF)

**S14 Fig. Comparison of protein expression differences between atria and ventricle with other studies.** Protein abundance ratios between atria and ventricle as reported in this study (x-axis) were compared to 2 independent studies reporting chamber-specific protein expression in human hearts (y-axis). Top: comparison to Linscheid et al. (2020) [14]. Bottom: comparison to Doll et al. (2017) [39]. Proteins deemed significant in either atria (blue) or ventricle (red) in our study were highlighted, and agreement on higher expression in the same chamber was determined for each study as annotated. Analyses are based on data presented in S9 Table.
(TIF)

**S15 Fig. Fold change protein intensity between atria and ventricle compared to independent datasets.** Fold changes from Fig 4D were compared with independent chamber-specific proteome datasets. Human protein expression fold changes were compared against fold changes derived from Doll et al. (2017) [39] and Linscheid et al. (2020) [14]. Analyses are based on data presented in S9 Table.
(TIFF)

**S16 Fig. Profile plots of protein expression of disease-related genes in left ventricle of all species measured.** Genes related to HCM are shown on the left and DCM on the right. Top row: absolute protein intensity on log10 scale. Faint lines depict single measurements, and points and thick lines show mean values. Bottom: ratio of protein intensity of each species relative to human, calculated from mean protein intensities, separately for each protein. Species abbreviations: H, horse; Hu, human; M, mouse; P, pig; R, rat; Z, zebrafish. Only proteins with no missing data points were included. Analyses are based on data presented in S8 Table. DCM, dilated cardiomyopathy; HCM, hypertrophic cardiomyopathy.
(TIF)

## Acknowledgments

The authors would like to thank the PRO-MS Danish National Mass Spectrometry Platform for Functional Proteomics and the CPR Mass Spectrometry Platform for instrument support and assistance.

## Author Contributions

**Conceptualization:** Alicia Lundby.

**Data curation:** Nora Linscheid.

**Formal analysis:** Nora Linscheid, Alberto Santos, Ulrike Leurs, Johan Z. Ye.

**Funding acquisition:** Alicia Lundby.

**Investigation:** Nora Linscheid, Pi Camilla Poulsen, Robert W. Mills, Alicia Lundby.

**Methodology:** Nora Linscheid, Alberto Santos, Pi Camilla Poulsen, Lars J. Jensen, Jesper V. Olsen, Alicia Lundby.

**Project administration:** Nora Linscheid, Alicia Lundby.

**Resources:** Kirstine Calloe, Morten B. Thomsen, Bo H. Bentzen, Pia R. Lundegaard, Morten S. Olesen, Jesper V. Olsen, Alicia Lundby.

**Supervision:** Lars J. Jensen, Alicia Lundby.

**Validation:** Nora Linscheid.

**Visualization:** Nora Linscheid, Christian Stolte, Lars J. Jensen.

**Writing – original draft:** Nora Linscheid, Alicia Lundby.

**Writing – review & editing:** Nora Linscheid, Robert W. Mills, Ulrike Leurs, Christian Stolte, Pia R. Lundegaard, Morten S. Olesen, Lars J. Jensen, Alicia Lundby.

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
