## [Editor Report · Decision Letter 0]

2 Sep 2020

Dear Dr Lundby, 

Thank you for submitting your manuscript entitled "Quantitative Proteome Resource across Hearts from Humans and Model Organisms" for consideration as a Research Article by PLOS Biology.

Your manuscript has now been evaluated by the PLOS Biology editorial staff as well as by an academic editor with relevant expertise and I am writing to let you know that we would like to send your submission out for external peer review.

Please re-submit your manuscript within two working days, i.e. by Sep 04 2020 11:59PM.

Kind regards,

Aaron Nicholas Bruns, Ph.D.,

Associate Editor

PLOS Biology

---

## [Decision Letter · Decision Letter 1]

22 Oct 2020

Dear Dr Lundby,

Thank you very much for submitting your manuscript "Quantitative Proteome Resource across Hearts from Humans and Model Organisms" for consideration as a Methods and Resources at PLOS Biology. Thank you also for your patience as we completed our editorial process, and please accept my apologies for the delay in providing you with our decision. Your manuscript has been evaluated by the PLOS Biology editors, an Academic Editor with relevant expertise, and by three independent reviewers.

As you will see, the reviewers agree that the scale of the study in terms of breadth of species and number of proteins analyzed is very impressive. Nevertheless Reviewer 2 thinks that the paper requires additional mechanistic insights or functional understanding in order to consider it further for publication in the journal. After consulting with the academic editor and the rest of the team, we have concluded that the resource presented in the study is invaluable for the research community and we do not think it is necessary to provide further mechanistic insights for this type of manuscript. However, all the other issues raised by the reviewers would need to be addressed.

In light of the reviews (attached below), we will not be able to accept the current version of the manuscript, but we would welcome re-submission of a revised version that takes into account the reviewers' comments. We cannot make any decision about publication until we have seen the revised manuscript and your response to the reviewers' comments. Your revised manuscript is also likely to be sent for further evaluation by the reviewers.

We expect to receive your revised manuscript within 3 months. 

**IMPORTANT - SUBMITTING YOUR REVISION**

*Re-submission Checklist*

*Published Peer Review*

*PLOS Data Policy*

*Blot and Gel Data Policy*

Sincerely,

Ines

--

Ines Alvarez-Garcia, PhD,

Senior Editor,

ialvarez-garcia@plos.org,

PLOS Biology

Reviewers’ comments

Rev. 1:

I was happy to review this manuscript by the Lundby group. The authors present a very comprehensive and detailed proteomic assessment of a number of cardiac tissues, representing the largest single proteomic study of its kind. They are to be commended on the depth and quality of this work. The repository that they have put together represents a very valuable addition to the field. I am particularly pleased to see the searchable online database: atlas.cardiacproteomics.com. [I did not search or query their associated PRIDE repository].

I had absolutely no concerns of the methods, protocols, interpretation, or discussion. In fact, I was not able to find even any spelling mistakes or errors! This is very surprising, but very refreshing!! I have not seen such a polished manuscript in over a decade and the authors have clearly invested a considerable amount of energy to get this paper into such top quality. As such, I am very happy to recommend publication of this manuscript.

Very minor queries at the scientific level.

I am not sure that Eggnogg is the most current approach for homology/paralogue searches. It is certainly acceptable, but I believe it is a little outdated perhaps. OrthoDb or orthognc may be a more current approach.

Fig 3C. What is meant by 'same regulation'? Would this not represent more of a similar "directionality". The word regulation has perhaps a different connotation in this setting.

Supplemental data figure S6D - what is the potential reasoning for the significant overlap of the PCA plots in the horse? There is considerable overlap /too much dispersion, particularly between the RA/LA but seems to not separate from the RV. This is contrast to all of the other tissues examined which showed very nice correlations (clear separations). Also in that the pearson correlations look very tight in panel b.

Overall I found this to be an outstanding publication and look forward to its acceptance.

Rev. 2:

Review of PBIOLOGY-D-20-02615R1 "Quantitative Proteome Resource across Hearts from Humans and Model Organisms" by Linscheid et al, submitted to PLOS Biology

Overview

Lundby and colleagues report an in depth, qualitative, and regiospecific mass spectrometry-based analysis of the cardiac proteomes of human in comparison to diverse animal models (pig, horse, rat, mouse and zebrafish). The breadth of coverage (~7,000 proteins) is commendable. Their subsequent comparative analysis reveals intriguing differences in protein expression between species (altered chamber specificity) and heart regions (left vs right ventricle vs atria), motivating possible caveats in using certain models to assess human disease, including leads associated with cardiac pathology that display pronounced model specific expression patterns, as well as suggesting candidates for future functional validation experiments. But in its current form, the study offers only limited descriptive value, lacks conceptual novelty, and falls short in terms of illuminating fundamental differences in biochemical function or pathogenesis, or cardiomyocyte specific expression.

Specific comments

Conceptual novelty - Recognizing the limitations of animal models of human disease is important, but it's not made clear how this work significantly advances understanding of the (human) cardiac proteome, which has been reported extensively on previously, as has that of mouse and rat. Hence, while this is a tour de force effort, that provides impressive coverage of the cardiac proteome across diverse organisms, it's not sufficiently clear how pathophysiologists will change their use of models based on these findings. This is compounded by the fact that the biological variance within (versus between) a species is not reported, which makes comparison across models potentially less meaningful.

The authors are encouraged to leverage their data sets and provide more compelling examples of mechanistic insights they have gleaned, for a specific model species or else human, as well as some form of independent validation experiments to verify certain key findings. The authors could better illustrate the functional relevance of the differences they observe, at least for a few select targets, as a proof of concept to more clearly demonstrate the novelty, mechanistic value and potential impact of the resource they have generated.

Rev. 3: Edward Lau - note that this reviewer has waived anonymity

Heart diseases are a major cause of morbidity and mortality in the world and multiple animal models are employed in biomedical researchers to identify potential disease mechanisms and translate results to human. The molecular differences between the heart of different model species await further investigations.

To address this, the authors compared the abundance of 7,000 proteins using label free mass spectrometry between human, pig, horse, rat, mouse, and zebrafish. Protein expressions are compared between left ventricles, right ventricles, left atriums and right atriums of the mammalian species, and between the atrium and ventricle for the two-chambered hearts in fish. They found:

* Protein profiles largely clustered by evolutionary lineages as expected

* Nevertheless, important differences in expression of major cardiac proteins (natriuretic peptides, myosin heavy and light chains, etc.) among model animals especially zebrafish

* Unexpected inversion of atrium/ventricle relative protein distributions across species

* Some species are more like human in their expression profiles of some disease-implicated proteins which may be taken into consideration when deciding on disease model.

Overall the looks like a competent study, applying state of the art mass spectrometry to survey of the proteins expressed in different chambers of the heart across five species. The proteomics experiments look to be well executed. There were some previous attempts to compare heart protein expression in different species which should be cited, but the current data set has more protein coverage and also compared different chambers. Some particular strengths of this dataset over previous proteomics comparisons of are the use of freshly collected samples, and resolving orthology in cross-species comparisons. Raw and summarized data are made publicly available for re-analysis, and the authors have created a web app for easy data interactions. I believe it will provide a useful resource for researchers in cardiac biology and human proteomes.

Even though this is primarily a resource study, I think there is an opportunity to provide additional contexts and interpretations for this large dataset. I have the following comments:

Major Comments:

1. There were previous attempts to compare the hearts of different species (e.g., Ref 14 and 15 cited in the manuscript, PMID 24070373, and potentially others). It might be reasonable to compare them to the current study (in terms of protein and species coverage) and also provide a short analysis of shared proteins and concordance or disagreements between the results.

2. How much of the protein expression differences across species might be due to differences in cell type proportion and distribution? In this regard the low loading of mitochondrial proteins across species is a notable results.

3. More generally, there isn't a lot of consideration given to why there are these species specific differences especially with regard to critical cardiac proteins. Which difference may be potentially attributable to differences in heart rate, metabolic needs, turnover, etc.?

4. Some statistics on animal model usage in heart research (e.g., PubMed trends and stats) might be helpful here. Is the horse a particular common model for heart biology as the authors suggest, compared to other animals like rabbits, etc.

5. Mapping proteins across species using an orthology database (EggNOG) is a strength and something not always done thoroughly in previous studies that compare species proteome. With the orthology data, can the authors estimate how much of the inter-species qualitative differences in major cardiac proteins are primarily "genomic" in origin (e.g., diversification after gene duplication etc.) and how much is exclusively "proteomic", e.g., all orthologs present in genome difference in tissue expression preference, etc.?

6. Model selection is often based on a variety of factors, including availability of reagents, ease of genetic manipulation, animal size for surgical model, costs, etc. It is not immediately clear where protein expression network comes in to play here or what is its relative importance. There is a tendency to gravitate toward common models and in so doing "miss out" on unique adaptations that can be informative. It would be more interesting if the authors could hypothesize based on the data what processes could the zebrafish/horse/etc. be a particularly good model for that we are not paying enough attention to.

7. The authors suggest that pigs make the best ARVC models based on the proteomics results, but mice are often used in ARVC studies. Can the data tell us anything on what might be the caveats there and how to potentially make new/better mouse ARVC models?

8. The inverted atrium/ventricle distribution is a surprising and remarkable results. Are there orthogonal lines of evidence that can corroborate this finding. In the human dataset here, is there a strong agreement with the relative A-V distribution of proteins compared with previous datasets (Human protein atlas, Doll et al. Nat Comm 2017, etc.)

Minor Comments:

1. Study limitations should be moved from supplementary methods to main text.

2. Figure 5a/b heat maps -- very difficult to see which value is missing due to similarity of color for NA values with the color scale.

---

## [Decision Letter · Decision Letter 2]

24 Jan 2021

Dear Dr Lundby,

Thank you for submitting your revised Methods and Resources entitled "Quantitative Proteome Resource across Hearts from Humans and Model Organisms" for publication in PLOS Biology. I have now obtained advice from two of the original reviewers and have discussed their comments with the Academic Editor. 

Based on the reviews, we will probably accept this manuscript for publication, assuming that you will modify the manuscript to address the data and other policy-related requests noted at the end of this email. In addition, we would like you to consider a suggestion to improve the title:

"Quantitative proteome comparison of human hearts with those of model organisms"

We expect to receive your revised manuscript within two weeks. Your revisions should address the specific points made by each reviewer. 

-  a cover letter that should detail your responses to any editorial requests.

*Published Peer Review History*

*Early Version*

Sincerely,

Ines

--

Ines Alvarez-Garcia, PhD,

Senior Editor,

PLOS Biology

ETHICS STATEMENT:

-- Thank you for including the name of the ethics committee for the approval of animal care. Please add the approval/protocol/license number.

--In addition, please add the ethics statements for both animal care and use of human tissue in the main Methods section of the manuscript.

Fig. 4A, B; Fig. 5C; Fig. S3A, D; Fig. S4A, D; Fig. S5A, D; Fig. S6A, D; Fig. S7A, D; Fig. S8A, D; Fig. S9A-C; Fig. S10; Fig. S11B and Fig. S13A, B; Fig. S14 and Fig. S16

Please also make the data deposited in the ProteomeXchange Consortium via the PRIDE(50) partner repository publicly available. Note that this is a requirement for the acceptance of the manuscript.

Reviewers’ comments

Rev. 2:

The authors have addressed most of the main concerns, and the revised paper is now suitable for publication.

Rev. 3:

Thank you for addressing my previous comments. I have no additional comments and look forward to seeing the manuscript published.

---

## [Editor Report · Decision Letter 3]

12 Feb 2021

Dear Dr Lundby,

On behalf of my colleagues and the Academic Editor, Cecilia Lo, I am pleased to say that we can in principle offer to publish your Methods and Resources entitled "Quantitative proteome comparison of human hearts with those of model organisms" in PLOS Biology, provided you address any remaining formatting and reporting issues. These will be detailed in an email that will follow this letter and that you will usually receive within 2-3 business days, during which time no action is required from you. Please note that we will not be able to formally accept your manuscript and schedule it for publication until you have made the required changes.

PRESS

Thank you again for supporting Open Access publishing. We look forward to publishing your paper in PLOS Biology. 

Sincerely, 

Ines

--

Ines Alvarez-Garcia, PhD 

Senior Editor 

PLOS Biology
